# Pre-Service University Training, Body Expression and Self-Concept

**DOI:** 10.3390/ijerph192316218

**Published:** 2022-12-04

**Authors:** Maria Rosario Romero-Martín, Daniel Caballero-Julia

**Affiliations:** 1Department of Musical, Plastic and Corporal Expression, Faculty of Health and Sports Sciences, University of Zaragoza, 22001 Huesca, Spain; 2Department of Musical, Plastic and Corporal Expression, Teacher Training School, University of Salamanca, 49029 Zamora, Spain

**Keywords:** Body Expression, self-concept, pre-service training, physical activity, Physical Education, COVID-19 pandemic, emotions

## Abstract

Body Expression (BE) has been defined in the past few decades as a discipline within Physical Education (PE) with very particular characteristics and a strong emotional component. In this study, a BE program was applied with university Physical Activity and Sports Sciences (PASS) students from six consecutive academic years: three prior to and three during the COVID-19 pandemic. A pre-post design was used to determine how the BE program affected the university students’ self-concept (SC). Thus, a questionnaire with a multidimensional approach to this construct was administered, with dimensions closely related to the BE program characteristics. The results revealed significant improvements in the final SC, compared to the initial SC. The men reported lower SC values than the women before the program’s implementation, but higher at the end. Therefore, the change was greater in the men, so the program may have had an equalizing effect between the groups. It was also verified that the pandemic had particularly affected the women.

## 1. Introduction

Educational goals have evolved throughout history from almost exclusive attention on cognitive outcomes to greater concern about social and emotional outcomes. Nowadays, emotion plays a predominant role. The theory of multiple intelligences [1], the advancement of neuroscience, and the findings of educational research from various areas of knowledge have focused on emotional factors in education. In the field of physical activity (PA), previous studies have confirmed that certain types of motor activities trigger different emotions, and have described how this affects the emotional intensity experienced by the subjects [2], as well as their effect on different aspects of human beings, such as self-concept (SC).

### 1.1. Body Expression: Motor Expression

Body Expression (BE) is a discipline that studies motor expression, which combines expressive, communicative and aesthetic aspects, as it has been defined in the last few decades in Spain. After a long period during which it was considered a ‘hotchpotch’ [3], and acknowledging that it continues to play a marginal role [4], its scope is now clearly defined and distinguished from other expressive disciplines, such as dance, mime, musical aerobic activities, aesthetic sports, etc., and its practice follows consistent internal logics.

Romero-Martín [5] defined it as the discipline that studies the organized forms of motor expression, based on a holistic body concept that promotes the creation of its own language through symbolization and creativity processes. The purpose of this practice is to express or externalize feelings or ideas, as well as to develop motor communication and movement aesthetics. Its aims are to learn skills and to discover body meanings, but also to improve well-being with one’s own and others’ bodies and to enhance personal development. Everything seems to indicate that BE is associated with the emotion generated during practice [6], contributing with its working techniques to the participants’ psycho-emotional balance [7] and helping to build personal identity and autonomy by means of symbolic play [8].

### 1.2. BE: Inhibition and Its Potential Factors in PASS Students

The nature of BE practice presents certain challenges in the educational context. Caballer, Oliver and Gil [9] verified in their study that teachers with experience in BE detected inhibition and refusal in the participants, which they attributed to the obstacles they encountered in their education and to self-imposed protecting blocks. All this hinders the natural expression of emotions and generates a superficial BE experience, especially in those participants with limited knowledge of their own bodies. Bara [10] referred to these blocks as BE mediators.

One of the reasons that may influence inhibition and refusal is the lack of skills frequently shown by participants during this type of practice. In physical activity-specific literature, possessing motor skills was identified with physical achievements associated with efficacy, usually related to sports skills. In this context, self-efficacy has been widely examined as a psychological variable. According to Bandura [11], it consists in trusting in your own abilities to organize and execute actions that will solve future situations. Various authors agreed that having high self-efficacy has a positive influence on motivation and determines the amount of effort and persistence that a person puts into the activity performed [12]. Similarly, competence has a high predictive function for being more physically active in PE classes [13]. In addition, it is shown that a feeling of competence reinforces SC [14]. In BE, the achievement lies in the ability to externalize or communicate feelings or ideas, and a low SC has a negative impact.

Additionally, the students of the Degree in Physical Activity and Sports Sciences (PASS), the population of the present study, usually have little experience in BE. In a wider context, the social imaginary of this degree is more associated with sports movement than with expressive or aesthetic movement, reflected in the fact that many students report little efficacy or skills in this field. This means that it is difficulty to change these perceptions, but this can be achieved mainly through determined support for the students to help improve their self-efficacy, as proposed by Bandura [15].

Another reason may be the identification of these activities with female stereotypes, so gender is revealed as an important variable when talking about BE. Related literature evidenced that motor expression activities are linked to gender stereotypes [16,17,18,19] because they involve behaviors that have traditionally been associated with women, such as plasticity, aesthetics or creativity. By contrast, they are inhibited when sports practice is proposed [20], as it is usually associated with male stereotypical characteristics: strength, endurance and motor skills. The identification with the female gender generates initial refusal in individuals who like sports, mostly men, who become a major methodological challenge for teachers. Moreover, the study by Romero-Martín [5] revealed that gender was one of the key factors of inhibition in BE. Likewise, Durán, Lavega, Sáenz de Ocáriz, Costes and Rodríguez [20] confirmed the influence of gender on inhibition.

All the above can be summarized in the cycle ‘fear + embarrassment + low skill perception + expectations → inhibition → refusal’, which teachers have to make an effort to counter(Figure 1).

Romero-Martín [5] analyzed the factors involved in this inhibition in university Physical Education students by examining the behaviors that hindered body expression and communication. Five factors that summarize the above were defined using qualitative techniques: (1) to be seen in public; (2) to be seen dancing in public; (3) physical contact due to physical, psychological or sex-role-related aspects; (4) to send messages that make us more well-known and vulnerable; and (5) lack of motor skills.

In short, the low self-perceived motor expression skills plus the embarrassment and fear of exposing their bodies and themselves due to multiple factors are part of a human being’s SC construct. All this allows us to describe a model to understand the emotional lattice of BE practice.

### 1.3. Positive Effects of BE Practice

As previously described, we are referring to a motor practice with high emotional content that produces intense affective reactions [21], which are sometimes negative (fear, embarrassment) [4], especially at the beginning of the programs [22].

Nevertheless, in addition to the emotional reactions mentioned, BE practice generates very positive experiences, reported by the students during and after the programs [4]. It was confirmed that it made the students feel good about themselves and with others [23], embarrassment decreased progressively and the level of social skills increased [24]. All these reactions produced personal satisfaction that affected SC as explained by Sonstroem, Weis, Sander, Sorensen, Stewart and Corbin (in Caballer, Oliver and Gil, [9], p. 2). Similarly, Sánchez López [25] observed a relationship between BE and certain SC aspects, as shown by Lavega in his extensive work. Various authors highlighted the importance of implementing this type of practice since it provides the participants with information that allows them to know themselves better [26], due to its emotional intensity.

Consequently, it is deemed important to include BE in the educational system since it contributes to the improvement of students’ SC [27]. Numerous studies based on the research by Shavelson, Hubner and Stanton [28] have presented evidence in all fields of the influence of SC on PA practice in the past few years. SC has been associated with performance, but also with the attitude (persistence or withdrawal) towards physical activity practice and towards expectations [15], which affect the student’s predisposition towards certain PA contents such as BE.

### 1.4. Model of Reciprocal Effects

The emotional lattice described influences upon SC just as SC determines the students’ emotional reactions in a bidirectional process, possibly similar to the ‘virtuous cycle’ described by Correa Romero et al. [29] (p. 174), composed of SC and other factors. This is not a unidirectional, but a reciprocal or bidirectional causal relationship, as proposed by Zulaika Isasti [30] for SC and academic performance.

Starting from this, we coincide with the model by Marsh, Papaioannou and Theodorakis [31], who showed in their study findings that support the model of reciprocal effects where SC can influence behavior in PE practice, and in turn, PE boosts the development of SC; however, studies like that by Garn et al. [32] report contradictory results in research on this reciprocal relationship.

### 1.5. Self-Concept, Multidimensionality

SC is considered to be one of the most determining variables in personality, especially from a motivational or emotional point of view [33].

Shavelson, Hubner and Stanton [28], in a major work, defined SC as a person’s perception of themself, which is “formed through their experience with their environment [...] and influenced especially by environmental reinforcements and significant others.” Thus, “one’s perceptions of oneself are thought to influence the way in which one acts, and one’s acts, in turn, influence the way in which one perceives oneself” (p. 411). Navajas Seco [27], based on Bandura, referred to SC as the global view someone builds of themself on the basis of their experience and the significant assessment others make of them and their behaviors. González-Pienda, Núñez Pérez, González-Pumariega and García García [33] defined it as the image someone has of themself depending on the integration of external and internal information, which is judged and assessed according to the individual’s reasoning style on the significant aspects of that information, with a strong emotional component. In short, whether we talk about perception, image or the global view of oneself, SC refers to the inner space an individual builds and rebuilds based on their own actions and how they make their perceptions of themself fit with the information received from the context.

*Multidimensionality.* There is plentiful specific literature devoted to the internal structure of SC. According to Pabago [34], there are two main models: one that understands SC as a global one-dimensional construct, and a multidimensional one composed of various structures that relate to different behavioral areas. Fernández-Zabala et al. [35] stated that multidimensionality had been broadly accepted since the 1970s, in agreement with Sanabrias-Moreno et al. [36].

Based on this multidimensional perspective, Shavelson et al. [28,32] developed a four-dimension model: academic, social, physical and emotional, which has been widely used by distinguished researchers, such as Harter [37], who focused on the educational aspect, and others. Starting from here, several authors have proposed other SC dimensions. For example, Goñi Grandmontagne, Ruiz de Azúa García and Rodríguez Fernández [38] mentioned five dimensions: general, general physical, physical skills, physical fitness and physical attractiveness; while Fernández-Zabala et al. [35] defined eleven dimensions: academic-verbal, academic-mathematical, physical skills, physical fitness, physical attractiveness, physical strength, honesty, emotional adjustment, autonomy, self-realization, social responsibility and social competence.

All this, and considering that a person’s SC cannot be seen but must be inferred [28], has led to the different models being crystalized in questionnaire proposals and research studies involving all or some of its dimensions (see Fernández-Zabala et al. [35], p. 14). In particular, the physical dimension has been widely analyzed in research about physical activity, the area of our study. One of the most well-known proposals was made by Fox and Corbin [39], consisting of a four-dimension model: sports competence, physical fitness, strength and physical attractiveness. This is the basis of the Physical Self-Perception Profile (PSPP) questionnaire, extensively adapted and published.

Nonetheless, various authors pointed out that multidimensionality needs to be adapted to the particular characteristics of the situation and they defended the importance of associating the concept with specific situations [28]. Therefore, the physical dimension of SC, as it has been previously defined by authors, does not completely address our study subject. BE cannot be exclusively explained by Fox and Corbin’s dimensions (sports competence, physical fitness, strength and physical attractiveness) or reduced to Blanco’s [40] model of motor competence and physical attractiveness. By contrast, dimensions focused on affective-emotional aspects should be taken into account to explain how BE, as we have defined it, affects SC. Consequently, we needed to search for more specific models and questionnaires that match the internal logic of the BE construct.

We found that the questionnaire proposed by Caballer, Oliver and Gil [9] contained dimensions that addressed aspects similar to the theoretical construct that is the basis of the curriculum of the university course in which this study was conducted. Therefore, this questionnaire was chosen to try to understand how PASS students’ SC changed after the implementation of a one-semester BE intervention program.

### 1.6. COVID-19 Pandemic and PA Practice

This research proposed a six-year longitudinal study to analyze how the COVID-19 pandemic may have affected students’ self-reported SC.

Starting in March 2020 and due to COVID-19, face to face lessons at Spanish universities were necessarily replaced by virtual sessions [41], including practical lessons. This was done thanks to the audiovisual means university and students put in place. In the following academic year (2020–2021), practical lessons were conducted on-site, with face masks and in half-sized groups that alternated every other week. In the academic year 2021–2022, face masks were used during most of the program and were removed towards the end. The pandemic seriously affected three academic years by dramatically changing students’ social routines [41]. This has been confirmed in previous studies, which reported anxiety, depression and stress among the population [42]. Cadena-Duarte [43] (p. 50) stated that these confinement-related health issues may alter the perception of physical SC and psychological well-being. This influenced the development of behavioral, social and affective skills that consolidate through the interpersonal interaction process [41] since the possibilities of having physical contact, sharing ideas among groups, or learning collaboratively were reduced or disappeared, as well as other emotional aspects that are heavily involved in body expression and communication. All in all, and with regard to SC, it seems logical to think that a change in the context conditions and between-subject habits would lead to a change in self-perception [34].

Due to all of the above, we present a study on self-concept in relation to motor expressive-communicative activity practice in order to gain knowledge on how this factor operates in PASS students. Consequently, the aims of this study were:To analyze the influence of a BE program on university PASS students’ SC, and to reveal which SC dimensions are more strongly affected and how.To study SC changes depending on gender.To examine the effect the COVID-19 pandemic may have had on SC in a specific university context.

## 2. Materials and Methods

### 2.1. Design

A single-group, pre-post [44], quasi-experimental [45] study was designed to compare the results from a questionnaire administered before and after a Body Expression intervention program, given the impossibility of controlling all the variables that influence SC as randomness is not possible when working with natural groups. It was a quantitative study, both cross-sectional, because two measurements were taken from the same group of students in one academic year, and longitudinal, because the data from six consecutive academic years were analyzed.

### 2.2. Population

The non-probabilistic sample was chosen for convenience and proximity [46]. The participants were the first-year students of the Degree in Physical Activity and Sports Sciences (PASS) from a Spanish public university who were enrolled in a Body Expression (BE) course and provided informed consent to participate in the study. Six consecutive academic years were examined: three before (2016–2017, 2017–2018 and 2018–2019) and three during (2019–2020, 2020–2021 and 2021–2022) the pandemic, with no students repeating the course.

The students who were enrolled and participated in the study are described in Table 1, divided by academic year, pandemic effect and gender.

### 2.3. Instruments

The questionnaire Scale for Body Expression/Communication for university students [9] was administered. The authors took the following steps to validate the questionnaire: (1) Theoretical concept definition; (2) Instrument analysis (SDQ III, STAI, ISRA, MMPI, EPQR and MPS were reviewed); (3) Assessment by professionals (expert panel system, with eight experts from four Spanish regions with at least eight years’ experience teaching BE at university); (4) First scale creation with 142 items divided into 6 factors; (5) Second analysis by the experts: the items with a between-expert agreement of 75% or above were accepted. As a result, the final scale was obtained. Subsequently, descriptive statistics and the corrected homogeneity index were calculated. In addition, internal consistency was determined through Cronbach’s alpha. The six dimensions yielded values between 0.74 and 0.87, all higher than 0.7, the value usually established as valid. The total questionnaire reliability was 0.87.

An 8-point Likert-type scale was used, where 1 meant ‘definitely false’ and 8 ‘definitely true’. Direct and reverse items were combined to prevent answering trends. Finally, the dimensions (factors) and the corresponding items were:(1)A = Physical appearance, referred to physical attractiveness regarding beauty or body structure, among others (items 1, 5, 13, 20 and 26);(2)C = Body expression/communication skills, BE-specific skills and contents (items 3, 6, 15, 21 and 28);(3)E = Emotional, emotional control aspects: anxiety, nervousness, optimism, depression, tension and level of concern about things (items 2, 8, 9, 22 and 25);(4)P = Problem-solving, individual’s creativity to combine ideas, enjoy and be interested in inventing new ways of solving problems, pleasure in imagination and originality (items 16, 18, 23, 24 and 29);(5)S = Relationships with people of the same sex, ability to establish relationships with people of the same sex (items 4, 7, 10, 12 and 19); and(6)O = Relationships with people of the opposite sex, ability to establish appropriate relationships with people of the opposite sex (items 11, 14, 17 and 27). The reverse items were: 2, 5, 8, 9, 11, 12, 17 and 26.

### 2.4. Intervention Program

A BE program was administered that focused on emotional aspects, which are key for the subject’s perception of SC, in line with the ideas expressed in the introduction so that the proposals of Zulaika Isasti [30] were taken into account for the improvement of SC. The program corresponded to a subject of the PASS degree, given in the second term of the first year. It consisted of 6 credits (60 face to face hours), distributed in 15 theoretical hours, 40 practical hours and 5 seminars in small groups. The first three years, all the classes were face to face. The sessions of the years affected by the pandemic combined face to face with remote classes. As the subject is given in the second term, the first year (2019–2020) began face to face and concluded as remote; and in the other two years, two groups were formed which alternated every other week.

The practical contents were: (1) Disinhibition and group building; (2) Body and self-knowledge; (3) Movement qualities; and (4) Feelings and sensitivity.

The methodological axes the course was based on were: (1) progression towards disinhibition; (2) progression from self-knowledge to expression and communication; (3) progression towards group integration and cooperation; (4) evolution from directed proposals to autonomous work and self-regulation; and (5) evolution from execution to creation capacity. Furthermore, the teaching intervention in class was focused on creating a space for expression [47] that was integrating, promoted respect, participation and acceptance of differences, and respected individual abilities. Affective feedback was used and students’ previous ideas and pre-conceptions about the topic were used as a basis. 

The types of activities carried out in the practical lessons included [5]: (1) Sensitivity and disinhibition (to foster the sense of belonging to the group, to lose fear or embarrassment upon contact, to show their non-sport or non-athletic motor skills, to generate an environment where individuals feel safe and confident to release their creativity and learn for personal development); (2) Becoming aware of the expressive, communicative and aesthetic possibilities of movement; (3) Representation and symbolization (ability to recreate or interpret feelings and ideas, either real or imaginary); (4) Movement qualities (discovery of the factors determining the motor expression action according to von Laban [48]; and (5) Acknowledgement of the gesture as a cultural fact [49].

Formative and continuous assessment was applied through peer-assessment, self-assessment and hetero-assessment techniques and using various means so that every student could be assessed according to their particularities. One means of assessment was the creation, performance and assessment of a group performance, which served as the reference social situation [50].

For the remote sessions, the teacher prepared an audiovisual document (Windows PowerPoint(Microsoft Office Professional Plus 365, Huesca, Spain)) with theoretical explanations, links to videos, articles or other materials, and proposals of physical practice that every student should individually perform and record. Then they would send their videos and pictures for the teacher and/or other students to review.

In order to show the adequacy of the questionnaire to the course curriculum, these two aspects are related in Table 2.

### 2.5. Variables

The independent variable was the Body Expression intervention program.

The dependent variables were SC and its six dimensions, considered as continuous variables: physical appearance, body expression/communication skills, emotional dimension, problem-solving ability, relationships with people of the same sex and relationships with people of the opposite sex.

The sociodemographic variables were gender and the COVID-19 influence. Two groups were created regarding the latter: academic years prior to the pandemic and academic years affected by the pandemic.

### 2.6. Procedure

The study began with the administration of the questionnaire proposed by Caballer, Oliver and Gil [9] at the beginning (February) of each of the six academic years examined (pre-measurement). Just before the beginning of the lessons, the students were sent a link to a Google form through the Moodle platform. On the form, they were advised that, if they accepted, it would be used for teaching and research purposes, and that it was anonymous, so it would not affect their marks. Subsequently, the Body Expression intervention program was implemented as previously described and the same questionnaire was administered again at the end (post-measurement) during the first days of June.

The set of goals, competencies and contents were maintained during the six academic years, and the methodology underwent usual basic, and also specific, adaptations in the years affected by the pandemic.

### 2.7. Data Analysis

A MANOVA Biplot [51], also known as a canonical biplot [52], was used for data analysis. This technique was designed in order to graphically represent the results of multivariate variance analysis in a reduced space. In this analysis, the dependent variables played the role of continuous variables, while the groups were the regression variables. MultBiplot software (Version 03/12/2018 (18.0312)) [53] was used. One-way and two-way analyses were applied in this study, depending on the SC pre- and post-measurement values, gender and the presence or not of the COVID-19 pandemic.

The graphical representation consisted of drawing the directions of maximum separation between groups on a bi-dimensional plane with the intention of revealing the differences between them, as well as the dependent variables (SC dimensions) that caused such differences. The graphical elements and their meaning were as follows:Circles: they represent every group of analysis determined by the variables. The centre is the mean (M) and the radius is the level of confidence estimated through a univariate test (SD).Vectors: they represent the variables (questionnaire dimensions); the arrowhead points in the direction of the maximum value. The angle defined between vectors is directly proportional to the correlation between variables. The vectors not only indicate the directions of maximum separation between groups, but also provide an estimation of the group mean values. Thus, groups (circles) located in the vector direction (the arrow pointing at positive variable values) present higher mean values than those groups that are close to the origin or even on the negative vector values.Distance between circles: the greater the distance between circles, the larger the difference between the groups they represent.Significant differences between groups: to calculate them, two circles are projected over the continuation of a specific variable (vector). If the circles do not overlap, this means that there are significant differences between the two groups in that variable.

## 3. Results

First, the results obtained from the one-way analysis will be presented, followed by the two-way analysis results. Due to space limitations, the general graphs will be shown and, in some cases, also particularly interesting projections.

### 3.1. One-Way Analysis

*Impact of the BE program on the SC, initial measurement vs. final measurement*. The MANOVA Biplot shown in Figure 2 reveals significant differences between the SC pre- and post-measurements. As can be seen, all the dimensions examined presented this difference, the post-measurements being higher than the pre-measurements.

*Gender.* Likewise, in general, significant differences were observed between the men and women (Figure 3). On the whole, the men presented higher values than the women, except for dimension 4 (Problem-solving), where the women scored higher. Nevertheless, after projecting the groups, these differences were only found to be significant at the 95% level in variables 1 (Physical appearance) and 5 (Relationships with the same sex).

*Pandemic*. With regard to the pandemic effect, Figure 4 reveals no significant differences between the academic years that were studied before and during the pandemic when they were analyzed without controlling for other factors.

### 3.2. Two-Way Analysis

In this section, we present several analyses that take into account the effect of the interaction of two factors on SC dimensions, as well as on the groups under study in this research.

By doing so, even though the one-way analysis did not yield significant differences between the groups in some cases, the combination of factors will help us better explain the behaviors observed.

*SC pre-* vs. *post-measurement, gender.*
Table 3 shows the details of the values of the descriptive analysis of the interaction between both variables. The means are between 5 and 6, on a scale of 8 points where 1 means “definitely false” and 8 means “definitely true”. The observation of the confidence limits leads us to consider significant differences at the 95% level.

New important data appeared in the interrelation of the variables, initial and final measurement, and gender. Significant differences were observed in the variables separately and also in the interrelation between them (Table 4). The effect size of the measured variable stands out (${\eta_{p}}^{2}$ = 0.266), indicating a high effect size. In the other two cases, they were moderate.

Figure 5 shows more graphically that:(1)The distance between the men pre-measurement and the men post-measurement (quadrants 1 and 2) was greater than between the women pre-measurement and the women post-measurement (quadrants 3 and 4).(2)The post-measurement values were significantly higher both in the men and women.(3)Consequently, the men’s post-measurement group presented the highest values.(4)The distance between the men and women was smaller in the post-measurement than in the pre-measurement.

(5)In the pre-measurement, the largest difference in SC between the men and women was in dimension 4 (Problem-solving) (Figure 6a), where the women presented higher values (as already seen in the one-way analysis). However, this difference completely disappeared in the post-measurement. By contrast, differences were found in the rest of the dimensions (Figure 6b), where they had not been observed before, with the men showing higher values.

*SC pre-* vs. *post-measurement, pandemic.*
Table 5 shows, in detail, the descriptive data of the groups derived from these variables. The means were between 5 and 6 on a scale of 8 points, so that they were above the median. 

The independent one-way analysis of the pandemic effect on students’ SC did not yield significant differences between the groups. Again, the effect size attributed to the measurement (Table 6) (impact of the BE program) was high (${\eta_{p}}^{2}$ = 0.378).

However, differences in the graphic result were found when the initial and final measurements were differentiated using the MANOVA Biplot.

As can be observed in Figure 7, the pre-measurement pre-pandemic and pre-measurement pandemic groups were clearly separated. In spite of this, no particular dimension was shown to have a differentiating effect. In other words, a global effect was observed, produced by the six SC dimensions.

Nonetheless, the overlap of the post-measurement projections indicated that there were no differences between them before and during the pandemic.

*Pandemic, gender.*Table 7 shows the descriptive data of the interrelation between the variables pandemic and gender.

Significant differences (*p*-value ≥ 0.05) are evident in the interaction of the variables pandemic and gender (Table 8). However, the effect size was small (${\eta_{p}}^{2}$ = 0.021).

When analyzing the figure obtained with the MANOVA Biplot (Figure 8), significant differences were found only in the women, who presented higher values in the academic years prior to the pandemic. This can be concluded from the distance observed between the circles representing women (quadrants 2 and 3) and from the location of the pre-pandemic women’s group, closer to the higher values (arrowheads) of the different vectors.

When looking for the dimensions causing such differences in the women, we found that the group projections on variables 2 (Body expression/communication skills), 4 (Problem-solving) and 6 (Relationships with the opposite sex) did not overlap, which can be interpreted as the presence of significant differences in these dimensions between the two periods.

## 4. Discussion

### 4.1. Summary of the Results

As regards study Aims 1 and 2, the results revealed that the SC values were higher after the program than prior to it, due to a joint effect of the SC dimensions. These improvements occurred both in the men and women.

There were also differences regarding gender. These differences were initially very weak but, when distinguishing between the pre- and post-measurements, it was clearly observed that the women presented higher SC than the men prior to the program’s implementation, while the opposite occurred after its completion, meaning that the men experienced a greater change.

Furthermore, the men’s and women’s post-measurement values were closer to each other than the pre-measurement ones, so the program may have had an equalizing effect on the SC between genders.

As regards the dimensions, a joint effect was initially observed, with D1 and D5 standing out to a small degree. Later, D4 was shown to play a relevant role, since it caused the higher value in the women at the beginning of the program and it was the only one not contributing to the men’s improvement at the end.

With regard to Aim 3, related to the pandemic effect, the pre-pandemic groups showed higher SC values than the pandemic groups at the beginning of the program. However, after the program, no differences were observed between the groups, which may be because the BE program had an equalizing effect on the groups’ SC. Furthermore, these initial differences appeared because of the women’s results, so it can be stated that the difference between before and during the pandemic was a gender issue. The pandemic particularly affected the women.

Based on the above summary of the results, they will now be discussed by aim.

### 4.2. Aim 1: SC Evolution

As regards SC, the study outcomes are deemed to be very positive, considering that significant improvements were detected in the whole sample, with a global effect observed in the dimensions after the BE program implementation.

In light of the results, the BE program may have contributed to the SC improvement shown by students, possibly due to the important presence of affective and emotional aspects that are considered to be SC factors [31]. Zulaika Isasti [30] stated that it is difficult to change SC if the experiences of the subjects contradict their self-perceived image. In any case, changes occur in certain aspects, but not in the SC in general.

These results are in line with previous studies involving BE, which analyzed emotional factors that, as has been shown, affect SC. Reigal [54] found improvements in state of mind thanks to BE work. Ruano [24] found that, after the implementation of a BE program in university students, embarrassment decreased and social skills improved both in men and women, with greater changes in the latter. Durán et al. [20] confirmed that the use of psychomotor expression situations led to a reduction in negative mood states, and sometimes, an increase in the positive vigor-activity state. Rodríguez Barquero and Araya Vargas [55] observed significant differences in the variable inhibition-empathy-distrust in students like those from our study. Likewise, another study with university students revealed an increase in self-confidence and self-esteem after a BE program [55].

If providing an emotional environment is key to allowing feelings and disinhibition to emerge in order to improve SC, then BE seems to be an especially advantageous activity. Romero-Martín, Gelpi Fleta, Mateu Serra, and Lavega Burgués [56] verified it in their study with PASS students, where motor expression and games triggered the most intense positive emotions, confirming that the variable type of activity predicted the relevance of the emotional experience to the student. Thus, we can say that the implemented BE program probably had a positive influence on SC.

### 4.3. Aim 2: Improvements Depending on Gender and Their Causes

In this study, both the men and women showed improvements. At the beginning of the program, the men reported lower SC values than the women due to the influence of dimension 4. By contrast, their values were higher at the end because the rest of the dimensions increased. Therefore, the improvement was larger in the men.

One reason why the men presented lower values at the beginning may be the little inclination of this group towards expressive content, since these activities are identified with female gender stereotypes such as plasticity, aesthetics or creativity, while male stereotype characteristics are strength, resistance or motor skills [20]. Various authors have discussed students’ content preferences depending on gender. Alvariñas Villaverde, Fernández Villarino and López-Villar [16] described how women identified with rhythmic and expressive physical activities. Blández, Fernández-García and Sierra [17] had previously reported that men’s and women’s preferences regarding sports practice were coming closer, although this was due to the greater interest of women in traditionally masculine sports and not the other way around.

However, the men presented higher values than the women at the end of the study, showing a greater progression. This may be due to a lower level at the beginning because of various reasons, such as limited previous experience in BE [5], lower inclination towards strongly affective contents, or low perceived self-efficacy [11] in this type of motor task, aspects that may affect SC. As a consequence of all of the above, the BE experience based on emotional tasks and respecting the student’s timing may have caused a novelty effect [57] on the men, due to the presence of previously unexplored skills, which has been mentioned by several authors to explain the motivation to practice physical activity.

The values observed in the men and women were closer at the end of the program. This may be because the BE program had an equalizing effect on the SC dimensions examined, which respond to the internal logic of the BE content [58]. In particular, the data revealed that this coming together was the consequence of the women’s increase in dimension 4 (Problem-solving) and the men’s improvement in the rest of the dimensions. Dimension 4 is clearly related to tasks oriented to the creation of choreographies and performances, which were recurrent in the BE program, similarly to Rial Rebullido and Villanueva’s [59] study. This suggests a greater development of these skills in the women after the program. Meanwhile, the men reported significantly higher values in the other five dimensions: physical appearance (D1), expression and communication skills (D2), emotional control (D3) and the ability to build relationships with people of the same sex (D5) or the opposite sex (D6). Altogether, this revealed a greater effect of the program on the male gender. If we consider the emotional aspects as SC factors, our findings are in keeping with the studies by Lavega [4,18,21], where men reported higher emotional intensity in general, and also in line with the Esnaola Etxaniz [60] study, where men scored higher than women after the intervention. Besides, it was also confirmed that women did not stand out in dimension 3 (Emotional) despite the identification of the female stereotype with these aspects.

This study, therefore, agrees with that of Pena Garrido and Repetto Talavera [61], who suggested that, in order to avoid the effects of the regression towards the mean, independent studies by gender should be conducted, and also with Lavega, Lagardera, March, Rovira, and Araújo [4], who recommended controlling for the variable gender, since socio-emotional effects were different in men and women.

### 4.4. Aim 3: Pandemic

Our study revealed that the pre-pandemic groups presented higher SC values than the pandemic groups at the beginning of the program. So, it is likely that students were affected by the confinement situation and restrictions. These results may be due to the presence of anxiety, depression and reaction to stress in the general population [42], which particularly affected adolescents (the group to which the participants in this study belonged), who showed anxiety and depression symptoms affecting their emotional development and self-concept [41]. Such an effect on SC was also observed in Cachón et al.’s [36] systematic review, where several studies mentioned a decline in self-esteem during the confinement period; and in the [43] study, where the participants, unavoidably isolated due to COVID-19, reported lower values in this psychological variable, partly because of the changes in all their routines (social, family, work, etc.). Similarly, [62] conclude that confinement emergencies deteriorate the perception of physical SC and psychological well-being, especially affecting university students due to academic demands.

Furthermore, these initial differences appeared because of the women’s results, so it can be stated that the difference between before and during the pandemic was a gender issue. The pandemic particularly affected women. Therefore, our study revealed a gender gap, bringing to light that the women affected by the pandemic faced the BE courses with a lower SC than those who did in the years prior to the pandemic. This agrees with Rodríguez-Rey, Garrido-Hernansaiz and Collado [63], whose study showed that 36.6% of the participants presented psychological distress, the psychological effect being stronger in youth and women. This proves the large psychological impact [63] (p. 550) of COVID-19, which has been unequal among different population groups.

Nonetheless, the pre-pandemic and pandemic groups reported similar values after the program’s completion. This may be due to an equalizing effect of the BE program on SC, considering the importance of the program’s affective factors and other characteristics on SC.

## 5. Conclusions

Regarding the first aim referring to the possible influence of a BE program on SC, the study showed an improvement in all the dimensions of SC. This could be due to the BE program given to the PASS university students as it was focused on affective-emotional aspects that influence the perception of SC. Moreover, the different groups obtained similar final values in SC, so that it is considered that the program had been able to exert an equalizing effect as it was centered on personal development and emotional wellbeing.

With respect to the second objective, gender was shown to be a determinant variable in the study of SC, with the fourth dimension relative to problem-solving that made the difference, so it is concluded that it would be appropriate to include gender as a differentiating aspect in studies on this topic.

The third aim referred to the effect that the COVID-19 pandemic could have had on SC in a specific university context. It was shown that the pandemic particularly affected the women, as those women affected by the COVID-19 pandemic began the program with lower SC values that those of the pre-pandemic group, although this evened out after the program.

After the study, it was concluded that the dimensions of SC in the questionnaire by Caballer, Oliver and Gil [9], are coherent with the internal logic of the SC, which suggests continuing to investigate the relations of both constructs using this tool. It was also considered that, after this study and the analyzed background, a model of SC related to BE can be envisioned, where the affective variables: fear, embarrassment, perception of expressive skills and expectations, generate a process of inhibition, which triggers the rejection of expressive motor activities, a question that gives rise to the need to continue to investigate in this line. Both topics mark the lines of future research.

Some limitations in the study suggest the need to continue to investigate this topic; for example, including control groups to increase the reliability of the results; differentiating the results of the three years affected by the pandemic, as there are subtleties among the three years that have not been studied; delving further into the causes of the initial differences among the pre-pandemic and pandemic groups, or between men and women, using qualitative studies; looking at the particularities of expressive motor activities related to other physical activities that also involve a strong emotional impact in the subjects; or studying the impact on each sex of carrying out physical activities associated with a stereotype of the other gender in university and other populations.

## Figures and Tables

**Figure 1 ijerph-19-16218-f001:**
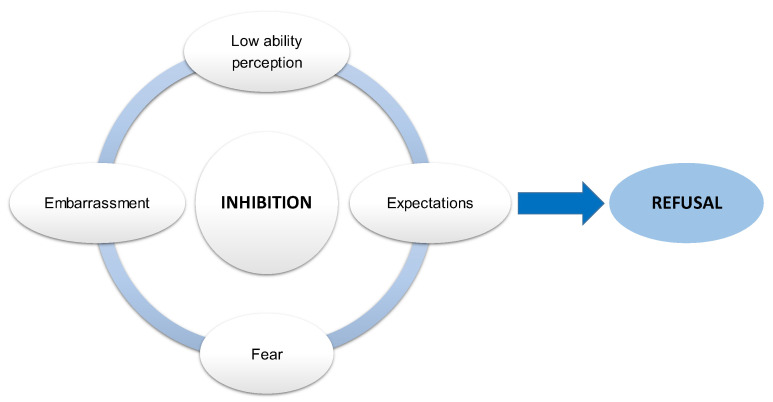
Proposed factor model that may explain students’ initial behavior.

**Figure 2 ijerph-19-16218-f002:**
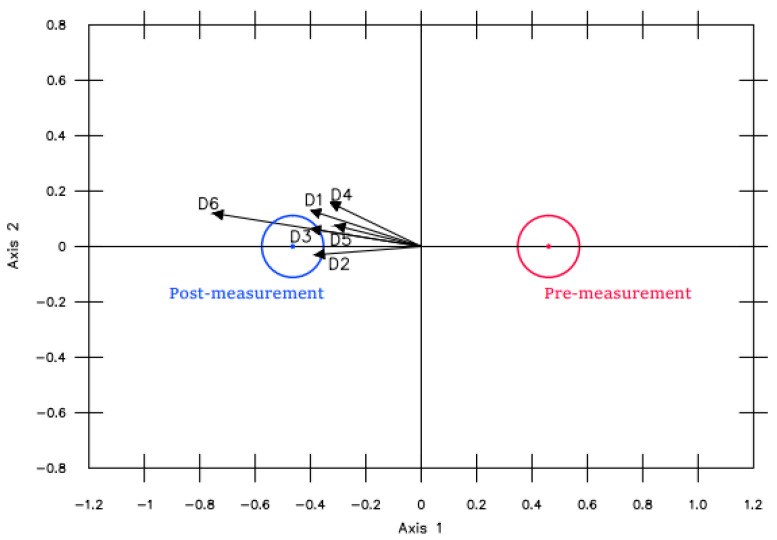
MANOVA Biplot pre- vs. post-measurement.

**Figure 3 ijerph-19-16218-f003:**
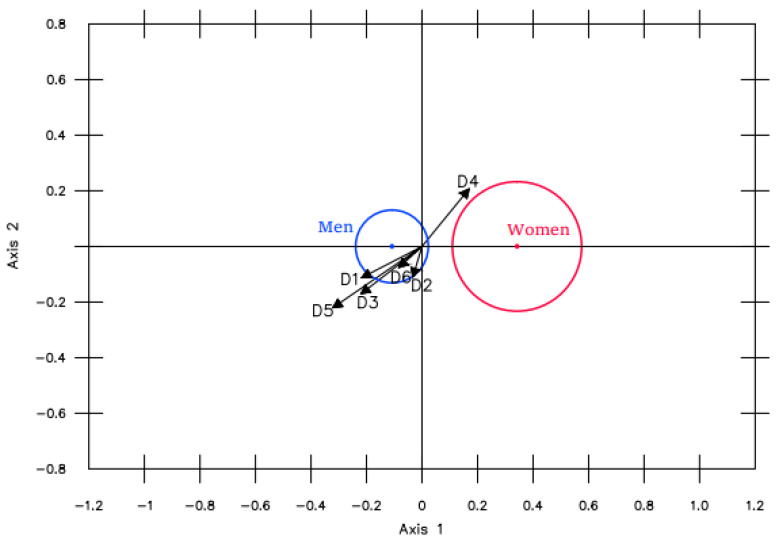
MANOVA Biplot comparing men and women.

**Figure 4 ijerph-19-16218-f004:**
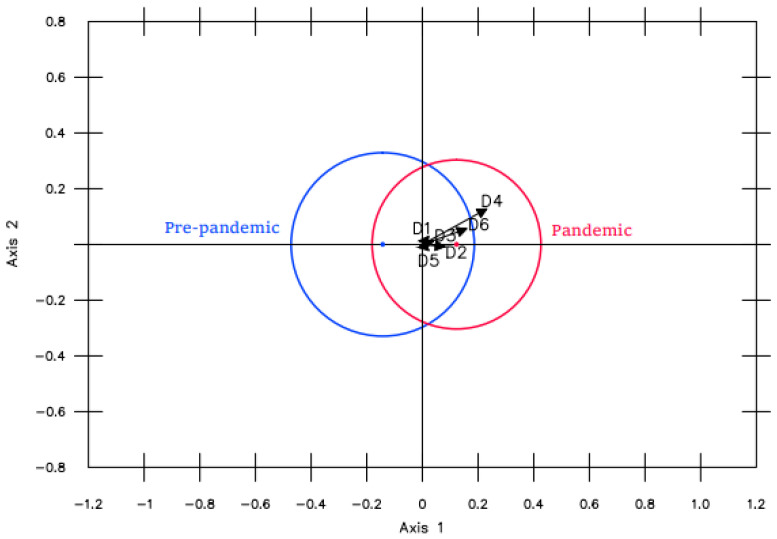
MANOVA Biplot comparing pre- and post-pandemic.

**Figure 5 ijerph-19-16218-f005:**
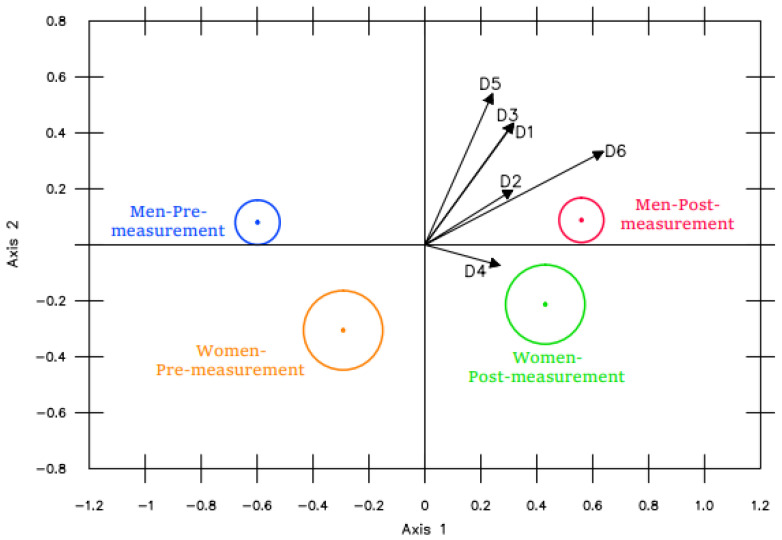
Men/Women + Pre-/Post-measurement.

**Figure 6 ijerph-19-16218-f006:**
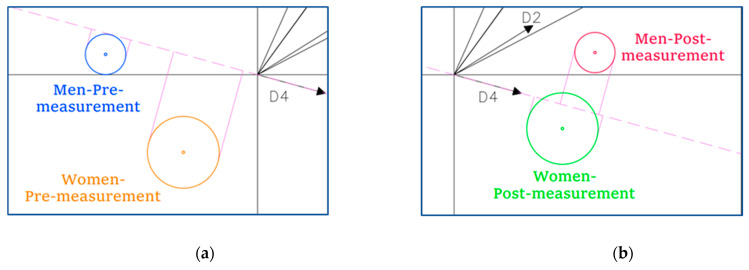
(**a**) *Zoom* PRE-measurements on D4; (**b**) *Zoom* POST-measurements on D4.

**Figure 7 ijerph-19-16218-f007:**
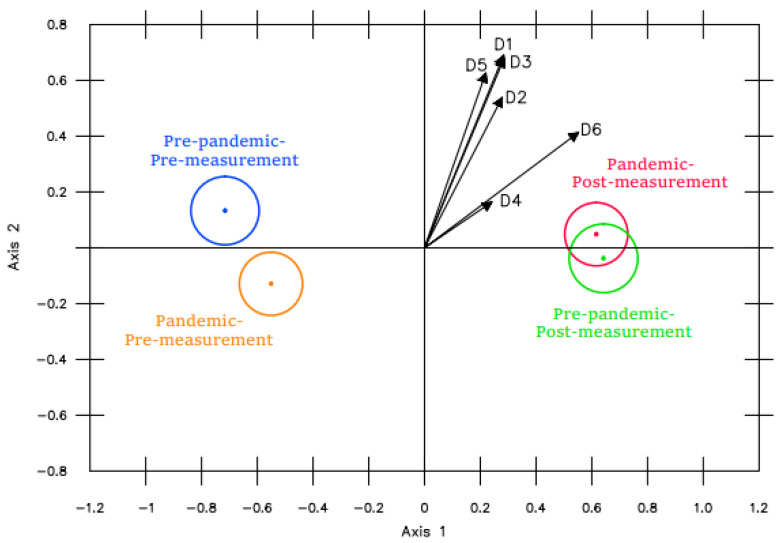
*Zoom* POST-measurements on D4.

**Figure 8 ijerph-19-16218-f008:**
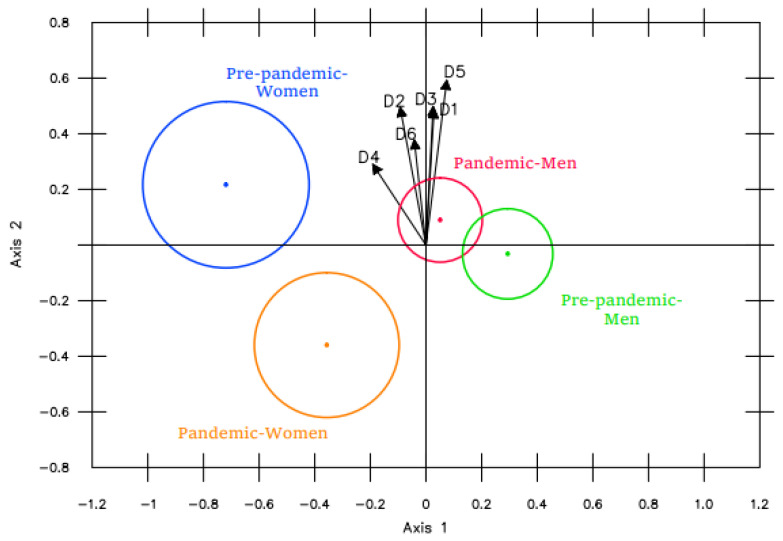
Before/During the pandemic + Pre-/Post-measurement.

**Table 1 ijerph-19-16218-t001:** Students participating in the study.

Year	Academic Year	Affected by the Pandemic	Participants				
			% Women	Average Age	% Men	Average Age	N
1	16–17	no	23.5	20.3	76.5	19.3	51
2	17–18	no	28.6	20.5	71.4	20.3	56
3	18–19	no	16.4	20.3	83.6	19.3	61
4	19–20	yes	25.4	20.0	74.6	19.6	63
5	20–21	yes	23.9	20.7	76.1	19.9	67
6	21–22	yes	26.5	20.1	73.5	20.0	68
TOTAL			24		76		366

**Table 2 ijerph-19-16218-t002:** Correspondence between the questionnaire dimensions and the intervention program.

Dimension	Intervention Program Aspects
A = Physical appearance, referred to physical attractiveness regarding beauty or body structure, among other aspects.	A specific BE aspect is addressed in the theoretical lessons: body models and own body image acceptance. With regard to the content block ‘Static body’, form composition is analyzed and self-perception and acceptance of one’s own body are introduced; a question which is one of the methodological lines of the course.
C = Body expression/communication skills, BE-specific skills and contents.	Developed through all learning activities.
E = Emotional, emotional control aspects: anxiety, nervousness, optimism, depression, tension and level of concern about things.	One content block is feelings and sensitivity. We work on basic emotions that are represented (acting) and felt (own feelings).
P = Problem-solving, individual’s creativity to combine ideas, enjoy and be interested in inventing new ways of solving problems, pleasure in imagination and originality.	Preparing sequences and performances, both in groups and individually. Use of learning situations through discovery methodologies during practical lessons.
S = Relationships with people of the same sex, ability to establish relationships with people of the same sex	Given the low number of women and the different types of groups used in class, women are more likely to work with people of the opposite sex than men. In the sessions, the groups should be mixed to try to adjust the imbalance between the numbers of both sexes.
O = Relationships with people of the opposite sex, ability to establish appropriate relationships with people of the opposite sex.	

**Table 3 ijerph-19-16218-t003:** Descriptive data of the initial–final measurements vs. gender.

Dimension	Gender	Measurement	Mean	SD	Standard Error	Confidence Interval at 95%	
						Lower Limit	Upper Limit
D1	Men	Pre	5.412	0.542	0.033	5.348	5.476
		Post	5.586	0.546	0.033	5.522	5.650
	Women	Pre	5.296	0.580	0.058	5.183	5.410
		Post	5.594	0.503	0.058	5.480	5.708
D2	Men	Pre	5.415	0.568	0.034	5.349	5.482
		Post	5.589	0.568	0.034	5.523	5.656
	Women	Pre	5.343	0.599	0.060	5.225	5.461
		Post	5.644	0.500	0.060	5.526	5.762
D3	Men	Pre	5.484	0.582	0.035	5.415	5.554
		Post	5.674	0.593	0.035	5.605	5.743
	Women	Pre	5.362	0.633	0.063	5.239	5.485
		Post	5.678	0.540	0.063	5.555	5.801
D4	Men	Pre	5.302	0.649	0.038	5.226	5.377
		Post	5.457	0.643	0.038	5.381	5.532
	Women	Pre	5.264	0.683	0.068	5.130	5.398
		Post	5.593	0.560	0.068	5.459	5.727
D5	Men	Pre	5.744	0.573	0.035	5.675	5.813
		Post	5.883	0.597	0.035	5.814	5.952
	Women	Pre	5.595	0.614	0.063	5.473	5.718
		Post	5.861	0.570	0.063	5.738	5.984
D6	Men	Pre	5.361	0.602	0.037	5.289	5.433
		Post	5.808	0.618	0.037	5.736	5.880
	Women	Pre	5.324	0.649	0.065	5.196	5.452
		Post	5.796	0.582	0.065	5.668	5.924

**Table 4 ijerph-19-16218-t004:** Significance values and effect size of the initial–final vs. gender values.

Effect	F	Hypothesis df	Error df	Sig.	${\eta_{p}}^{2}$
Measurement	43.644	6.000	723.000	0.000	0.266
Gender	6.121	6.000	723.000	0.000	0.048
Measurement vs. Gender	3.440	6.000	723.000	0.002	0.028

**Table 5 ijerph-19-16218-t005:** Descriptive data of the variables initial–final values vs. pandemic.

Dimension	COVID	Measurement	Mean	SD	Standard Error	Confidence Interval at 95%	
						Lower Limit	Upper Limit
D1	Pre-pandemic	Pre	5.417	0.5006	0.042	5.335	5.500
		Post	5.551	0.5395	0.042	5.469	5.634
	Pandemic	Pre	5.356	0.5927	0.039	5.280	5.432
		Post	5.618	0.5309	0.039	5.543	5.694
D2	Pre-pandemic	Pre	5.420	0.5255	0.044	5.335	5.505
		Post	5.567	0.5551	0.044	5.482	5.653
	Pandemic	Pre	5.379	0.6150	0.040	5.301	5.458
		Post	5.632	0.5499	0.040	5.553	5.711
D3	Pre-pandemic	Pre	5.489	0.5411	0.045	5.400	5.578
		Post	5.634	0.5856	0.045	5.545	5.723
	Pandemic	Pre	5.426	0.6393	0.042	5.344	5.508
		Post	5.709	0.5748	0.042	5.627	5.791
D4	Pre-pandemic	Pre	5.284	0.6130	0.050	5.186	5.381
		Post	5.456	0.6399	0.050	5.359	5.553
	Pandemic	Pre	5.300	0.6924	0.046	5.211	5.390
		Post	5.518	0.6141	0.046	5.429	5.608
D5	Pre-pandemic	Pre	5.743	0.5326	0.045	5.654	5.832
		Post	5.844	0.5908	0.045	5.755	5.933
	Pandemic	Pre	5.678	0.6277	0.042	5.596	5.760
		Post	5.906	0.5890	0.042	5.824	5.988
D6	Pre-pandemic	Pre	5.351	0.5700	0.047	5.259	5.444
		Post	5.777	0.6155	0.047	5.684	5.870
	Pandemic	Pre	5.353	0.6482	0.043	5.267	5.438
		Post	5.829	0.6030	0.043	5.744	5.914

**Table 6 ijerph-19-16218-t006:** Significance values and effect size of the variables of the initial–final measurements vs. the pandemic.

Effect	F	Hypothesis df	Error df	Sig.	${\eta_{p}}^{2}$
Pandemic	0.834	6.000	723.000	0.544	0.007
Measurement	73.307	6.000	723.000	0.000	0.378
Pandemic vs. Measurement	1.884	6.000	723.000	0.081	0.015

**Table 7 ijerph-19-16218-t007:** Descriptive data of the variables pandemic vs. gender.

Dimension	Pandemic	Gender	Mean	SD	Standard Error	Confidence Interval at 95%	
						Lower Limit	Upper Limit
D1	Pre-pandemic	Men	5.473	0.5230	0.034	5.406	5.541
		Women	5.521	0.5289	0.063	5.397	5.646
	Pandemic	Men	5.521	0.5728	0.032	5.458	5.584
		Women	5.387	0.5807	0.055	5.279	5.496
D2	Pre-pandemic	Men	5.466	0.5464	0.035	5.396	5.536
		Women	5.588	0.5316	0.066	5.459	5.717
	Pandemic	Men	5.534	0.5964	0.033	5.469	5.599
		Women	5.422	0.5908	0.057	5.309	5.534
D3	Pre-pandemic	Men	5.549	0.5673	0.037	5.476	5.622
		Women	5.604	0.5705	0.069	5.469	5.738
	Pandemic	Men	5.605	0.6180	0.035	5.537	5.673
		Women	5.457	0.6295	0.060	5.339	5.574
D4	Pre-pandemic	Men	5.332	0.6330	0.040	5.253	5.411
		Women	5.499	0.6131	0.074	5.353	5.645
	Pandemic	Men	5.421	0.6627	0.038	5.347	5.495
		Women	5.375	0.6648	0.065	5.248	5.502
D5	Pre-pandemic	Men	5.785	0.5576	0.037	5.713	5.857
		Women	5.824	0.5878	0.068	5.691	5.958
	Pandemic	Men	5.838	0.6150	0.034	5.771	5.906
		Women	5.655	0.6112	0.059	5.539	5.771
D6	Pre-pandemic	Men	5.545	0.6222	0.040	5.466	5.624
		Women	5.629	0.6538	0.075	5.483	5.776
	Pandemic	Men	5.619	0.6709	0.038	5.545	5.693
		Women	5.507	0.6603	0.065	5.380	5.635

**Table 8 ijerph-19-16218-t008:** Significance values and effect size of the variables gender vs. pandemic.

Effect	F	Hypothesis df	Error df	Sig.	${\eta_{p}}^{2}$
Género	6.368	6.000	723.000	0.000	0.050
Pandemic	1.008	6.000	723.000	0.419	0.008
Género vs. Pandemic	2.631	6.000	723.000	0.016	0.021

## Data Availability

Not applicable.

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
