# Peer review of "Pre-Service University Training, Body Expression and Self-Concept"

_ijerph, 2022, doi:10.3390/ijerph192316218_

Round 1

Reviewer 1 Report

Dear authors, 

First of all, congratulations on your work. Below are some suggestions that may improve the quality of your contribution:

- There are several parts of the document that are not translated, so it is suggested that they be revised.  

- Regarding Figure 1, the explanation of the model in the text is clear, so I suggest that it be removed.

- Regarding the research design, it can be seen that it may be a pre-experimental design. Adequate justification is suggested.

- In Table 1, it is suggested that the decimal notation "," be changed to ".".

- Regarding the study sample, it is suggested that the mean age of the sample be included, and that the inclusion-exclusion requirements (for example, has it been checked that no remedial students who have already taken the subject before are included in the sample).

- With regard to the items that make up each of the dimensions, both in the Physical appearance and Body expression/communication skills dimensions, item 26 is repeated.

- In relation to Table 2, which details the correspondence between the dimensions of the questionnaire and the intervention programme, with respect to dimensions M and O, reference is made to the limitation found during the implementation of the programme. In this respect, it is recommended to detail, as in the rest of the dimensions, which aspects of the intervention programme have dealt with these issues and how the limitation of unequal sex ratio has been solved or reduced.    

- Initially, on reading the document, it appears that a specific programme has been implemented to improve the SC of PASS, although on review it appears to be based on taking the subject of Body Expression. In this respect, the objective is to acquire a series of competences established in the national legislation in force, the purpose of which is to train professionals in physical activity and sport, and must therefore be adapted to the syllabus established beforehand in the subject, so that a specific programme to improve the SC of this population is not developed. Therefore, it is suggested that the authors justify the suitability of the programme.

- Regarding the remote sessions detailed, at the beginning of the explanation of the programme it is stated that the programme consists of 60 face-to-face hours. It is suggested that the authors provide more information in this regard and whether these were carried out during the entire programme or only in specific periods motivated by COVID.

- Similarly, although information is provided in the introduction about the adaptations that were carried out in the national territory due to the COVID, it is recommended to detail when the pre-test and post-test were carried out in order to really know in what situation the students were in at that time and if this could have affected the results. 

- The figures provided in the results are of great help for their interpretation. However, it is suggested that the authors provide in a table the data corresponding to the MANOVA results as well as another with the most relevant descriptive data according to gender and COVID, as well as the p-value and the effect size.

- With respect to the references, the concordance between how authors are cited in the document and subsequently in the references should be checked, adopting the same format when authors with two surnames are involved (for example number 17: in text Romero and Romero-Martín in references). 

- To facilitate the reader's understanding, it is recommended that each objective be clearly associated with its conclusion. In this section it would also be advisable to go into more detail on future lines of research and to detail the limitations of the study.

- The references are mostly from articles written in Spanish and, on some occasions, published a long time ago. Therefore, in order to improve the quality of the document, it is recommended that the authors search for articles in other languages that have been published in the last five years. These can reinforce the statements in both the introduction and the discussion.

Reviewer 2 Report

The article addresses an interesting topic and presents the results clearly.

However, the authors should also consider the following suggestions:

1. Introduction

- Line 91: Authors should submit Figure 1 in English.

2. Materials and Methods

- Line 218: Section 2.2. Population should be presented in English.

- How were your participants chosen? (Were these students and the University PASS chosen according to a certain criterion? The authors should mention what were the criteria / reasons for choosing this University and these students and not others).

- For all Tables and Figures inserted in the text, I would recommend the authors to mention the source.

- Line 261 - 262: It seems that this idea has already been written in English above.

4. Discussion

- Line 517 - 518: It is not necessary to mention the initials of the first names of these authors.

5. Conclusion

- The authors should highlight the implications of the findings on all stakeholders (university, students, researchers, others). Also, perhaps the contributions (implications of this study) should be grouped into theoretical and practical contributions.

-The authors should also point out the limitations and future directions of this research.

Round 2

Reviewer 1 Report

Dear authors, 

Congratulations on the improvement of the scientific paper. In this regard, I believe that the current version can be proposed for publication.